# A Comparative Analysis of Chemical, Plasma and In Situ Modification of Graphene Nanoplateletes for Improved Performance of Fused Filament Fabricated Thermoplastic Polyurethane Composites Parts

**DOI:** 10.3390/polym14235182

**Published:** 2022-11-28

**Authors:** Xiaojie Zhang, Jianhua Xiao, Jinkuk Kim, Lan Cao

**Affiliations:** 1School of Materials Science Engineering, Nanchang Hangkong University, Nanchang 330000, China; 2School of Chemistry and Chemical Engineering, Shanghai University of Engineering Science, Shanghai 201620, China; 3Department of Materials Engineering and Convergence Technology, Engineering Research Institute, Gyeongsang National University, Jinju 52828, Republic of Korea; 4School of Polymer Science and Engineering, Qingdao University of Science and Technology, Qingdao 266042, China

**Keywords:** FDM, graphene nanoplatelets, low-temperature plasma, surface modification

## Abstract

The limited number of materials and mechanical weakness of fused deposition modeling (FDM) parts are deficiencies of FDM technology. The preparation of polymer composites parts with suitable filler is a promising method to improve the properties of the 3D printed parts. However, the agglomerate of filler makes its difficult disperse in the matrix. In this work, graphene nanoplatelets (GnPs) were surface modified with chemical, low-temperature plasma and in situ methods, in order to apply them as fillers for thermoplastic polyurethane (TPU). Following its modification, the surface chemical composition of GnPs was analyzed. Three wt% of surface-modified GnPs were incorporated into TPU to produce FDM filaments using a melting compounding process. Their effects on rheology properties and electrical conductivity on TPU/GnPs composites, as well as the dimensional accuracy and mechanical properties of FDM parts, are compared. The images of sample facture surfaces were examined by scanning electron microscope (SEM) to determine the dispersion of GnPs. Results indicate that chemical treatment of GnPs with zwitterionic surfactant is a good candidate to significantly enhance TPU filaments, when considering the FDM parts demonstrated the highest mechanical properties and lowest dimensional accuracy.

## 1. Introduction

Three-dimensional (3D) printing is revolutionizing different important industrial areas, as it is distinguished from traditional processing techniques by its ability to manufacture components from prototypes to complex geometries with great design flexibility, high recyclability and less material waste [1,2]. Among the different 3D printing techniques, fused deposition modeling (FDM) has been widely used because of its easy handling, relative inexpensiveness and low chemical toxicity [3].

Although there is abundant availability of FDM machines, in many cases, FDM processes lack in providing consistency and reliability in terms of part properties, accuracy and finish, which limits the widespread application of FDM. These deficiencies depend mainly on the materials used, process parameters and post-processing techniques. Several studies and reviews of the literature have confirmed the effectiveness of optimizing the geometry, operation-specific parameters and annealing treatment for improving the manufactured parts [4,5,6,7]. With regard to the materials, till now, only a few commercial thermoplastic filaments have been available for FDM techniques, such polylactic acid (PLA), polyamide (Nylon), acrylonitrile butadiene styrene (ABS), thermoplastic polyurethane (TPU), polyetheretherketone (PEEK), etc. [8]. High-performance materials are highly demanded due to the versatility of FDM applications, such as engineering, automotive composites and aviation fields. The incorporation of fillers such as glass fiber, carbon fiber, ceramic or carbon-based nano-size are common approaches to attain this goal [9].

In recent years, graphene and its derivatives have received considerable attention due to their excellent properties. However, low-cost, high-quality and eco-friendly processes for manufacturing graphene are still challenging [10]. Compared with graphene oxide and other graphene-related materials, graphene nanoplatelets (GnPs) can be easily produced in large scale with physical methods and are commercially available with different particle sizes at a relatively low cost and have been identified as a substitute for graphene. GnPs exhibit exciting properties such as light weight and high electrical conductivity and mechanical strength. Polymer/GnPs composites exhibit more efficient improvement in terms of strength and thermal stability as compared with reduced graphene oxide (rGO), graphene oxide (GO), and multiwalled carbon nanotubes (MWCNT) [11,12]. Consequently, polymer/GnPs nanocomposites are popular and widely researched for many applications including 3D printing materials, low-cost composites films, wearable devices, medical hydrogels and many more [13]. Masarra and Batistella et al. [14] fabricated an electrically conductive circuit using PLA/PLC/GnPs by FDM. Misra and Ostadhossein et al. [15] constructed a multidrug-eluting stent using direct 3D printing from polycaprolactone/GnPs biodegradable composites. Jing and Chen et al. [16] prepared high thermal conductive polyethylene/GnPs nanocomposites for heat diffusion application of some advanced electronic devices. Li et al. [17] FDM printed a wearable pressure sensor based on TPU/GnPs nanocomposites. However, the platelet structure of GnPs exhibits some disadvantages. During the preparation of nanocomposites, GnPs tend to agglomerate, because of a considerable extent of π–π stacking interactions and their weak interactions with the polymer chains [18]. Agglomerates decrease the surface area of GnPs and, correspondingly, deteriorate reinforcement performance [19,20,21]. Attempts have been made to develop the dispersion of GnPs in nanocomposites, including utilizing chemical and physical surfactants modification, covalent functionalization and low-temperature plasma modification [18,22,23,24].

Based on the above, this work aims to investigate a series of approaches to treat GnPs, including chemical, low-temperature plasma and in situ modification, providing a comparative overview of the effect of surface modification on the quality of FDM-printed TPU/GnPs, in terms of dimensional accuracy and mechanical performance.

## 2. Materials and Methods

### 2.1. Materials

Thermoplastic polyurethane (TPU), LM-95A, was purchased from The Lubrizol Corporation (Wickliffe, OH, USA). Graphene nanoplatelets (GnPs), HGP-10, were supplied by Qingdao Yanhai Carbon Materials Co., Ltd. (Qingdao, China), with the average diameter around 8 µm and 1~15 nm thickness of graphene stacked into GnPs particle. 12-aminododecanoic acid (ADA), 98% purity, was bought from Shanghai Macklin Biochemical Co., Ltd. (Shanghai, China). Dimethylformamide (DMF), 97% purity, was purchased from Xilong Scientific Co., Ltd. (Shantou, China).

### 2.2. GnPs Surface Modification

#### 2.2.1. Low-Temperature Plasma Modification (PGnPs)

Plasma treatment was conducted in a plasma cleaner, model JS-P200, Hefei Jieshuo Vacuum Technology (Hefei, China). GnPs were exposed to air plasma for 50 min under powder condition of 100 W. During the treatment, the gas flow and gas low rate were kept as 40 Pa and 5 sccm.

#### 2.2.2. Chemical Modification of GnPs (CGnPs)

The GnPs and the zwitterionic surfactant ADA (GnPs:ADA = 10:1, weight ratio) were dispersed in DMF followed by stirring overnight at 85 °C, then the dispersion was sonicated for 3 h using an ultrasonic cleaner (SB-3200DTD, Ningbo Scientz Biotechnology, Ningbo, China) at frequency of 37 kHz. After that, the GnPs was separated from the suspension and washed by the DMF. Finally, the GnPs was dried for 48 h under vacuum at 90 °C [13].

#### 2.2.3. In Situ Modification (IGnPs)

The GnPs were premixed with the ADA prior to the extrusion process. The ratio of ADA to GnPs was 1:10 (wt/wt). Then, melt mixing of TPU and the GnPs/ADA hybrids (3 wt%) composites were processed using a signal-screw extruder (Wuhan Yiyang Plastic Machinery, Wuhan, China). The screw speed was 25 rpm. Extruder barrel temperatures were set as follows: 195 °C, 190 °C and 185 °C.

### 2.3. Filament Fabrication

To produce the desired filament size (1.75 mm) for use in FDM 3D printer, a twin-screw extruder (SHJ-20C, Nanjing Giant Machinery, Nanjing, China) was used. To ensure a homogenous dispersion of 3% GnPs into TPU, the TPU and GnPs were stir-mixed prior to melt compounding. The mixture was then fed into the extruder. The recommended starting extrusion temperature of TPU is 138 °C. To solve the die swelling problems and increase the dispersion of GnPs in the matrix, the barrel temperatures were set at 170 °C/180 °C/190 °C/185 °C/180 °C/170 °C, in order. The filament was subsequently cooled and then wound onto a reel.

### 2.4. Dog-Bone Specimens Printed by FDM

The preparation of the filament and the FDM printing process are shown in Figure 1. Initially, the FDM parts were modeled on CAD software according to the standard of ISO-527-2-2012 type-1BA with 5 mm thickness and exported in STL format. The STL file was sliced and transformed into G-code file by slicing software (Ultimaker Cura Version 4.2, Shenzhen, China). The dog-bone FDM parts were prepared by a commercial desktop FDM 3D printer (4Max Pro2.0, Shenzhen Anycubic, Shenzhen, China). In order to focus on the effect of GnPs, 3D printing process parameters such as layer thickness, feed rate, fill pattern, fill percentage or temperature were fixed for all the samples. Table 1 shows the values of these parameters. After printing, the supports were removed, and no further post-processing process was required.

### 2.5. Characterization

#### 2.5.1. Fourier Transfer Infrared (FTIR) Spectroscopy

Fourier transfer infrared (FTIR) spectra of modified GnPs were collected in the wavenumber range of 400–4000 cm^−1^ with a VERTEX70 (Bruker, Ettlingen, Germany) spectrometer. The samples were prepared by KBr-disk method [25,26].

#### 2.5.2. Rheological Properties

A capillary rheometer (MLW-400B, Changchun Intelligent Instrument and Equipment, Changchun, China) was used to measure the rheological properties, having a die of 1 mm diameter and 10 mm length. The apparent viscosity of the blends was determined in the shear rates ranging from 100 to 1500 s^−1^ at 190 °C [27,28].

The melt flow index (MFI) [29,30] was carried out through the XNR-400C melt flow indexer (Jinhe Instruments, Chengde, China) at 190 °C by applying a 3.24 kg load to extrude the molten polymers. A 100 g rod was used a plunger. The MFI values were generated from at least five determinations.

#### 2.5.3. Electrical Conductivity Measurement

The electrical conductivity was measured by using a resistance tester (AT683, Applent Instruments, Changzhou, China). The electrical conductivity (σ, S/m) was calculated using the following Equation (1) [31]:σ = 1/ρ = L/RS(1)
where ρ is the resistivity and R is the resistance. S and L are the length and cross-sectional area of the filament.

#### 2.5.4. Dimensional Accuracy

Dimensions of the 3D-printed dog-bone specimens in length direction were measured using a digital Vernier caliper and compared with CAD dimensions. The dimensional deviations in length were calculated for different TPU/GnPs-composite-based dog-bone specimens and compared with CAD dimensions.

#### 2.5.5. Surface Roughness

The average surface roughness (Ra) of the top surface of FDM parts is measured with a surface roughness tester (TR150A, Timech, Beijing, China) at 1 mm/s tracing speed, tracing length at 6 mm.

#### 2.5.6. Tensile Properties

Tensile strength and elongation at break were conducted using a universal testing machine, YF-900 (Yuanfeng, Yangzhou, China) based on the ISO 527-1-2021, at a crosshead speed of 200 mm/min, and the sample length between benchmarks was 50 mm. At least five specimens were tested for each composite, and the medial values were reported.

#### 2.5.7. Morphological Properties

The fracture surface morphologies of composites were investigated by field emission scanning electron microscopy (FE-SEM, FEI Nova NanoSEM 450, Brno, Czech Republic). The GnPs and facture surfaces of various TPU/GnPs filaments were sputtered with a thin layer of aurum [32].

## 3. Results

### 3.1. GnPs Characterization

FTIR was applied to analyze changes in chemical groups on the surface of GnPs before and after surface modification. Figure 2a presents the FTIR spectra of the GnP, PGnPs and CGnPs. The FTIR spectra of GnPs without any treatment shows bands at 2921–2765 cm^−1^ resulting from the C–H stretching. The amount 1151–1074 cm^−1^ corresponds to the C-O bending, and the C–H bending assigned at 1398, 860, 771 cm^−1^ is observed. Comparing pristine GnPs and PGnPs spectra, the low-temperature plasma treatment grafts’ various functional polar groups on the GnPs surface, including O–H, N–H or NH_2_ stretching ranging from 3056 to 3689 cm^−1^, C=O stretching at 1747 cm^−1^ and C–N or C–O stretching at 1000–1294 cm^−1^ wavenumber, the existence of the characteristic bands confirmed the functional polar groups that contain oxygen or nitrogen on the surface of GnPs after plasma treatment (Figure 2b). The polar groups on PGnPs surface will benefit the homogenous dispersion of GnPs in the matrix and the interfacial adhesion between filler and macromolecules. Noticeably, in the FTIR spectra of CGnPs, the additional peaks at 3451, 1380 cm^−1^ and 1741 cm^−1^ are assigned to O–H bending and C=O stretching of the carboxylic groups. The peaks at 3451, 1639 and 1461 cm^−1^ ascribe to the N–H bending and stretching of the amine group, while the broad brands at 1226–1025 cm^−1^ are the uptake for C–N. From the results of the FTIR spectra, we conclude that the modification of GnPs by ADA was successful (Figure 2b) [33,34,35,36,37,38,39]. The organophilic absorption between the aliphatic chain of ADD and the TPU matrix leads to uniform dispersion of particles in the TPU matrix.

### 3.2. Rheological Properties

Rheological properties can confirm the molecular entanglement and molecular relaxation of the polymer composites and guideline of the final construction of FDM parts [40,41]. Figure 3a,b demonstrate the relationship between shear stress and shear viscosity on the steady shear rate of the different TPU/GnPs composites. All composites display the non-Newtonian characteristics; the shear viscosity of all TPU/GnPs composites decrease sharply with increasing shear rate, showing shear thinning behavior. As revealed in Figure 3a,b, at low shear rate, in the range of 100–800 s^−1^, the shear stresses and shear viscosities of TPU/PGnPs and TPU/CGnPs are higher than those of TPU/GnPs. It is considered that PGnPs and CGnPs impede the chain mobility of TPU, during which their stresses and viscosities increase. However, the addition of IGnPs in TPU results in an obvious reduction in the shear stress and shear viscosity. This change indicates that the zwitterionic surfactant ADA is more of a plasticizer in the TPU/IGnPs composites than a surface modifier for GnPs during in situ treatment. Hence, the TPU/IGnPs filament shows a smoother surface than other filaments (Figure 3e). Over the entire range of shear rates investigated, all the non-Newtonian fluids followed a power-law relationship. At higher shear rates, the shear stress and shear viscosity of all the composites are revealed comparable regardless of the incorporation of fillers [41]. Figure 3c shows log–log plots of shear viscosity versus shear rate and the power-law index (*n*) for all TPU/GnPs composites at 190 °C. Comparing with TPU/GnPs composites (*n* = 0.39), both TPU/PGnPs (*n* = 0.22) and TPU/CGnPs (*n* = 0.31) composites have lower power-law index values, indicating that the surface modification of GnPs surface increases the interaction between TPU polymer chains and GnPs. The apparent viscosity of polymer melts is inversely proportional to their melt flow index (MFI). As shown in Figure 3d, the MFI of TPU/IGnPs increases slightly, while the TPU/PGnPs and TPU/CGnPs MFI values are smaller than that of TPU/GnPs. The variation is consistent with the shear viscosity. In summary, the four composites exhibit different viscosities (also different interactions between filler and polymer chains) in the order of TPU/IGnPs < TPU/GnPs < TPU/PGnPs < TPU/CGnPs [42,43].

### 3.3. Effect of GnPs on Electrical Conductivity of TPU/GnPs Composites

Figure 4 depicts the electrical conductivity of various TPU/GnPs composites. It is observed that the electrical conductivity was associated with surface modification of GnPs. However, these differences were not very significant or diverse in the same order of magnitude. The highest conductivity is achieved by TPU/CGnPs; the electrical conductivity increased by 196%. The functionalization of GnPs helps in the uniform dispersion of GnPs flakes throughout the polymer matrix, forming an inter-connected network for electrical conduction. As a result, electrical properties will be increased [44,45].

### 3.4. Mechanical Properties of FDM Parts

The change in functional groups on the surface of GnPs can differ polymer–filler interactions in the respective composites. To confirm the attractive polymer–filler interaction in TPU and GnPs experimentally, the mechanical properties of the various TPU/GnPs composites were examined and compared, and the tensile strength and elongation at break of the different TPU/GnPs composites are presented in Figure 5a,b. A serious relation between the surface treatment of GnPs and mechanical properties of composites can clearly be observed. IGnPs reduced the tensile strength of the composites, but when using PGnPs and CGnPs, the increase was 22% and 23%, respectively, which could improve the structural application of the composites. This result indicates that the polymer–filler interactions between TPU and PGnPs or CGnPs are stronger than those between TPU and IGnPs, which agree with the results of the rheological properties of composites. Meanwhile, the elongation at break values of the surface-modified GnPs exhibits slightly more greatly than the virgin GnPs composites. An application of CGnPs significantly increases the tensile strength and elongation at the break of composites, i.e., up to 69.79 MPa and 645%, respectively [37].

### 3.5. Dimensional Accuracy

Table 2 summarizes the mean values and standard deviations of dimensional variation in length between the FDM-printed parts, the CAD dimensions and the top surface roughness of the prototypes. Specimens showed values of positive differences, meaning that the length value exceed the CAD files. TPU/CGnPs composites show the most dimensional accuracy, while TPU/PGnPs show the least dimensional accuracy. More specifically, TPU/CGnPs and TPU/PGnPs show a mean error of 1.96% and 2.58%, respectively, while TPU/GnPs and TPU/IGnPs depict a mean error of 2.52% and 2.12%, respectively. In short, surface modification of GnPs can affect the dimensional accuracy of TPU/GnPs composites specimens; moreover, TPU/CGnPs samples show the best dimensional performance, probably because of the high viscosity of TPU/CGnPs. High viscosity allows for an increase in the content of spaces between paths [46,47,48,49,50]. Table 2 also indicates that homogenous distribution of modified GnPs in TPU helps in enhancement of the surface roughness. However, all specimens have high roughness values due to the layer-by-layer deposition process of the FDM technique.

### 3.6. Morphologies of Various TPU/GnPs Composites

Homogenous dispersion and distribution of the GnPs in the TPU matrix is a vertical factor for the property enhancement of the 3D-printed nanocomposites. The morphologies of pristine GnPs and the fracture surfaces of the various TPU/GnPs filaments characterized by FE-SEM are displayed in Figure 6. As Figure 6a shows, the GnPs is quite thin with smooth surface, and certain GnPs are aggregated and corrugated with a diameter of a few micrometers. The fracture surfaces of the TPU/GnPs (Figure 6b) and TPU/IGnPs (Figure 6c) filaments are significantly different from the fracture surfaces of TPU/GnPs (Figure 6d) and TPU/IGnPs (Figure 6e) filaments. Figure 6b,c shows rough and tortuous pathways, and the surfaces of the composites containing platelets projecting outside from the surfaces, meaning a weak adhesion to the matrix, and GnPs particles agglomerated into big agglomerates segregate in the TPU matrix. In contrast, the facture surfaces of the TPU/CGnPs filament (Figure 6d) are quite smooth and flat; the CGnPs seem to be well-embedded in the TPU matrix, and bundle formations gradually disappear for the composites, an indication of strong interfacial interactions between the TPU and CGnPs, which are in good agreement with the high mechanical properties of TPU/CGnPs composites [51,52].

## 4. Conclusions

In this work, the three different surface treatments for GnPs have been compared, and their effects on the rheological properties of TPU/GnPs composites, as well as the mechanical performance, dimensional accuracy and surface roughness of FDM-printed TPU/GnPs specimens, have been analyzed. Furthermore, microscope imaging has provided insight into the reasons behind the observed changes in mechanical performance. The introduction of IGnPs decreased the viscosity of TPU/IGnPs composites. There were no significant rheological differences between TPU/PGnPs and TPU/CGnPs composites, but TPU/CGnPs composites exhibited higher electrical conductivity. TPU/CGnPs FDM parts showed a significant improvement of the dimensional accuracy and mechanical properties over the other three materials, mainly because the stronger interfacial interactions between TPU and CGnPs. However, the PGnPs is more inclined to reduce the dimensional accuracy of FDM parts. The TPU/IGnP composites exhibited comparable dimensional accuracy compared with the TPU/CGnPs composites. However, it is also noted that the TPU/IGnPS composites demonstrated the lowest tensile strength among the four types of composites. All specimens present similar but high surface roughness due to the nature of FDM techniques; post-processing is necessary for further application. The results suggest that GnPs modified with ADA is more effective in improving the multifunctional properties of TPU.

## Figures and Tables

**Figure 1 polymers-14-05182-f001:**
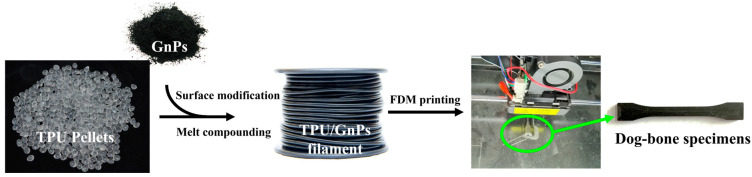
Preparation and FDM printing of specimens.

**Figure 2 polymers-14-05182-f002:**
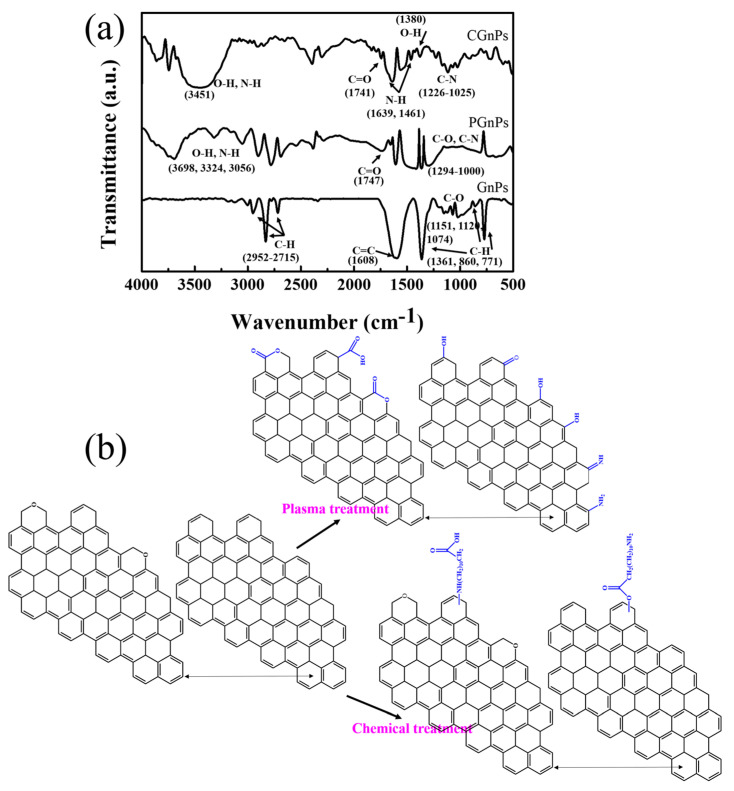
(**a**) FTIR spectra of pristine GnPs, PGnPs and CGnPs; (**b**) structure of pristine GnPs before and after surface modification.

**Figure 3 polymers-14-05182-f003:**
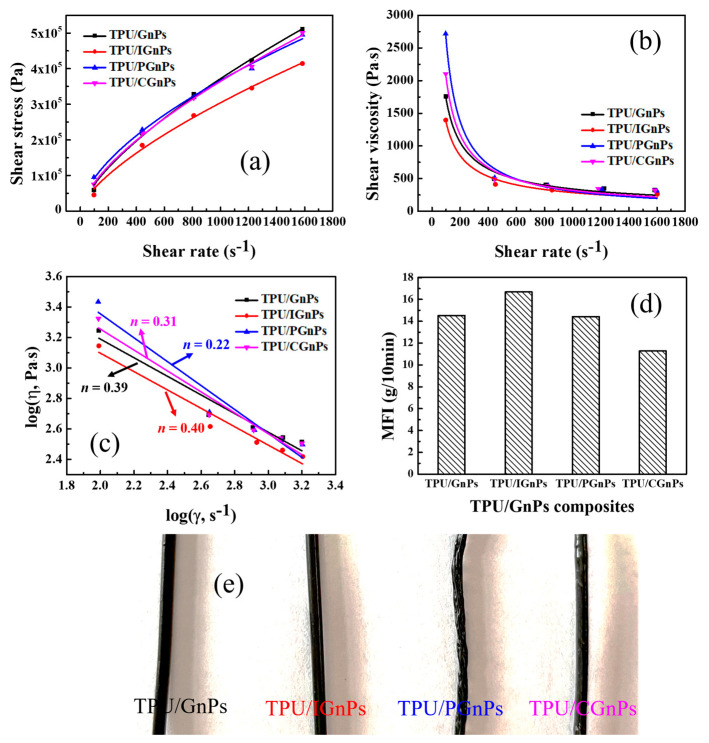
(**a**) Shear stress, (**b**) shear viscosity, (**c**) the bilogarithmic plots of viscosity versus shear rate, (**d**) MFI of various TPU/GnPs composites and (**e**) various TPU/GnPs-based filaments.

**Figure 4 polymers-14-05182-f004:**
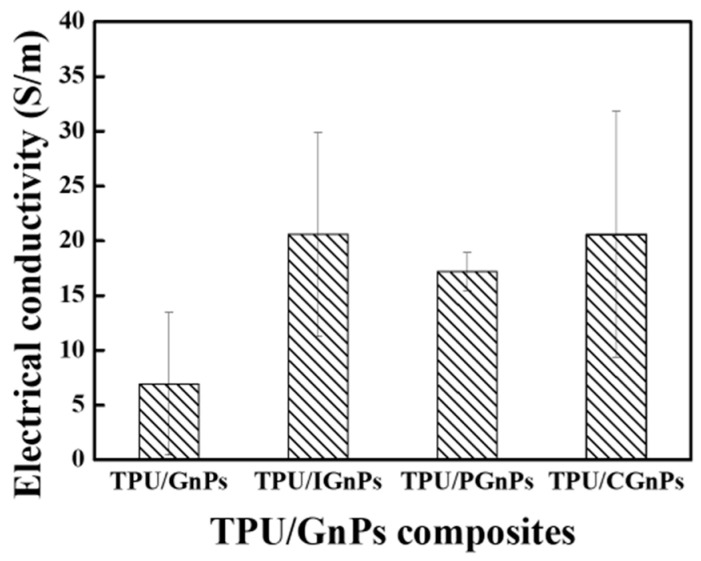
Electrical conductivity of various TPU/GnPs composites.

**Figure 5 polymers-14-05182-f005:**
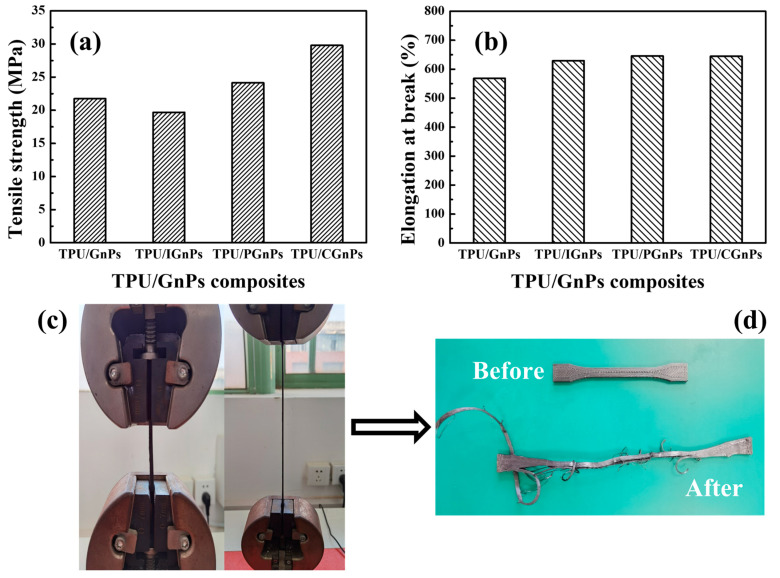
(**a**) Tensile strength and (**b**) elongation at break of various TPU/GnPs composites, (**c**) the deformed FDM parts at various stages and (**d**) the FDM-printed specimens before and after break.

**Figure 6 polymers-14-05182-f006:**
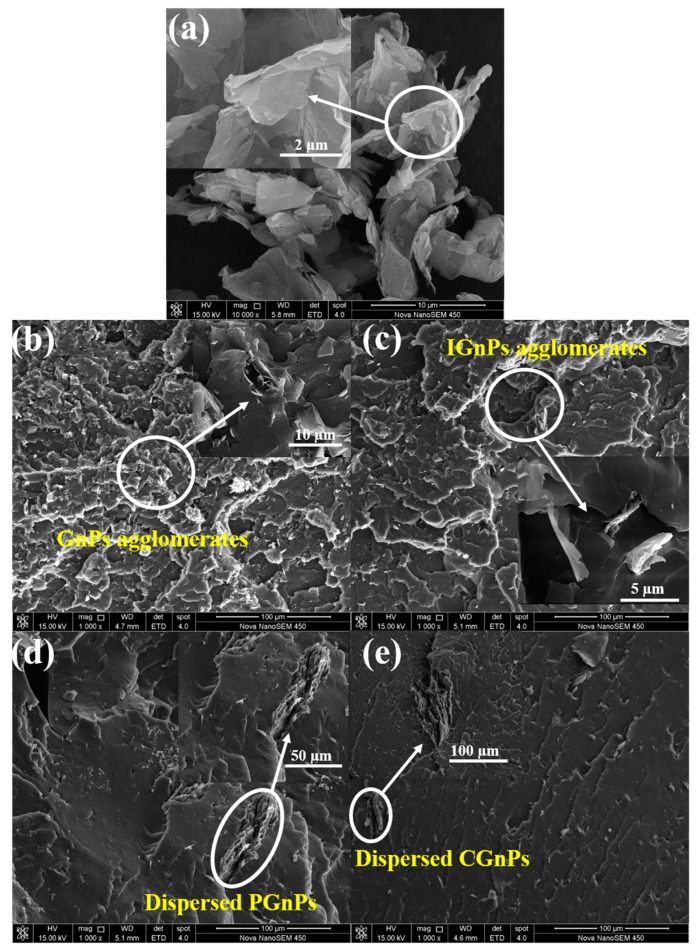
FE-SEM images of (**a**) GnPs and the fracture surfaces of (**b**) TPU/GnPs, (**c**) TPU/IGnPs, (**d**) TPU/PGnPs and (**e**) TPU/CGnPs filaments.

**Table 1 polymers-14-05182-t001:** 3D printer parameters.

Parameters	Value
Material	TPU-GnPs
Nozzle diameter	0.4 mm
Layer thickness	0.1 mm
Printing speed	40 mm/s
Nozzle temperature	190 °C
Bed Temperature	60 °C
Top/Bottom solid layers	1.2 mm
Outline/perimeters shell	1.2 mm
Internal fill pattern	Mesh
External fill pattern	Rectilinear
Internal fill percentage	20%
Filament diameter	1.75 mm

**Table 2 polymers-14-05182-t002:** Mean value and standard deviation of the dimensional deviation and surface roughness.

Property	Samples	Mean Value	Standard Deviation
Dimensional deviation (%)	TPU/GnPs	2.52	0.015
	TPU/IGnPs	2.12	0.108
	TPU/PGnPs	2.58	0.077
	TPU/CGnPs	1.96	0.081
Surface roughness (µm)	TPU/GnPs	1.98	0.384
	TPU/IGnPs	1.91	0.233
	TPU/PGnPs	1.78	0.265
	TPU/CGnPs	1.82	0.252

## Data Availability

The data presented in this study are available on request from the corresponding author.

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
