# Peer review of "A Comparative Analysis of Chemical, Plasma and In Situ Modification of Graphene Nanoplateletes for Improved Performance of Fused Filament Fabricated Thermoplastic Polyurethane Composites Parts"

_polymers, 2022, doi:10.3390/polym14235182_

Round 1

Reviewer 1 Report

      Dear Authors,   The article is interesting, but it requires a few changes, below is a list:   1. In the case of 3D printing, the basic mechanical properties are given the same in this article, and the important ones are omitted. Proposes to supplement the literature review for rheological analysis and known types of finishing processing, for example: a) Rheological Properties of Polyamide Pa 2200 in SLS Technology, doi10.17559/TV-2019025122204 b) Quality of the Surface Texture and Mechanical Properties of FDM Printed Samples After Thermal and Chemical Treatment, 10.5545/SV-JME.2019.6322   2. Chapter 2.3. Please include the temperature recommended by the manufacturer.   3. How many samples were made?   4. There are no 3D printer parameters and technical information about the production process.   5. The drawings are too small and its quality  need to be improved.   6. Recommendations to present the results of metrological measurements in the form of a table with average value, standard deviation etc and providing the parameters of production.   7. It seems that applications should be supplemented with analysis of the influence of processing into roughness, waviness or microstructure, if possible.   regards, Reviewer

Author Response

Dear Authors,  

The article is interesting, but it requires a few changes,

----------------------------------------------------------------------------------------

We thanks the referee for appreciating our article. As per the kind suggestion of the referee, we have tried our best to emphasize clarity in the revised version of our manuscript.

----------------------------------------------------------------------------------------

below is a list:  

  1. In the case of 3D printing, the basic mechanical properties are given the same in this article, and the important ones are omitted. Proposes to supplement the literature review for rheological analysis and known types of finishing processing, for example: a) Rheological Properties of Polyamide Pa 2200 in SLS Technology, doi10.17559/TV-2019025122204 b) Quality of the Surface Texture and Mechanical Properties of FDM Printed Samples After Thermal and Chemical Treatment, 10.5545/SV-JME.2019.6322

----------------------------------------------------------------------------------------

Sincerest thanks for supplying these articles to enrich our content.

According to the suggestion we have modified the literature review as follows:

Part I:

“Despite abundant availability of FDM machines, only a few suitable thermoplastic polymers are available for FDM technique such acrylonitrile butadiene styrene (ABS), polylactic acid (PLA), polyamide (Nylon), thermoplastic polyurethane (TPU), polyetheretherketone (PEEK) et al.”

modified as follow:

“Despite abundant availability of FDM machines, however, in many cases, FDM process lack in providing consistency and reliability in terms of parts properties, accuracy and finish, which limited the widespread application of FDM. These deficiencies depend mainly on the materials used, process parameters and post-processing techniques. Several studies and reviews of literature have confirmed the effectiveness of optimizing the geometry, operation specific parameters and annealing treatment for improving the manufactured parts.[1-4] With regard to the materials, till now, only a few commercial thermoplastic filaments are available for FDM technique such polylactic acid (PLA), polyamide (Nylon), acrylonitrile butadiene styrene (ABS), thermoplastic polyurethane (TPU), polyeth-eretherketone (PEEK) et al.”

Part II:

“ Rheological properties provide information of molecular entanglement and molecular relaxation of the polymer composites.”

modified as follow:

“ Rheological properties can confirm the molecular entanglement and molecular relaxation of the polymer composites, and guideline of the final construction of FDM parts.[5, 6]”

[1] T. Kozior, A. Mamun, M. Trabelsi, L. Sabantina, A. Ehrmann, Quality of the Surface Texture and Mechanical Properties of FDM Printed Samples after Thermal and Chemical Treatment, Strojniski Vestnik/Journal of Mechanical Engineering 66 (2020)

[2] A. Garg, A. Bhattacharya, A. Batish, Chemical vapor treatment of ABS parts built by FDM: Analysis of surface finish and mechanical strength, The International Journal of Advanced Manufacturing Technology 89 (2017) 2175-2191.

[3] S. Gao, R. Liu, H. Xin, H. Liang, Y. Wang, J. Jia, The Surface Characteristics, Microstructure and Mechanical Properties of PEEK Printed by Fused Deposition Modeling with Different Raster Angles, Polymers 14 (2021) 77.

[4] J.S. Chohan, R. Singh, Pre and post processing techniques to improve surface characteristics of FDM parts: a state of art review and future applications, Rapid Prototyping J. 23 (2017) 495-513.

[5] T. Kozior, Rheological properties of polyamide pa 2200 in sls technology, Tehnički vjesnik 27 (2020) 1092-1100.

[6] S. Thumsorn, W. Prasong, T. Kurose, A. Ishigami, Y. Kobayashi, H. Ito, Rheological Behavior and Dynamic Mechanical Properties for Interpretation of Layer Adhesion in FDM 3D Printing, Polymers 14 (2022) 2721.

----------------------------------------------------------------------------------------

  1. Chapter 2.3. Please include the temperature recommended by the manufacturer.

----------------------------------------------------------------------------------------------------------------------------

Thanks for the kind suggestion. The recommended starting extrusion temperature is 138 °C. To solve the die swelling problems and enhance the dispersion of GnPs in the matrix, we set the processing temperature as at 170 °C/180 °C/190 °C/185 °C/180 °C/170 °C.  In[7], the author manufactured the TPU filament in extruder using the same type of TPU at higher processing temperature, i.e.  190 °C/195 °C/200 °C/200 °C.

[7] J. Xiao, Y. Gao, The manufacture of 3D printing of medical grade TPU, Progress in Additive Manufacturing 2 (2017) 117-123.

----------------------------------------------------------------------------------------

  1. How many samples were made?

----------------------------------------------------------------------------------------

Each sample set consisted of five specimens, with a total of 20 specimens for testing.

----------------------------------------------------------------------------------------

  1. There are no 3D printer parameters and technical information about the production process.

----------------------------------------------------------------------------------------

Sincerest thanks for the suggestion. Initially, the FDM parts were modeled on CAD software according to the standard of ISO-527-2-2012 type-1BA with 5 mm thickness and exported in STL format. The STL file was sliced and transformed into G-code file by Ultimaker Cura 4.2 (slicing software). The dog bone FDM parts were prepared by feeding the extruded filaments into a commercial desktop FDM 3D printer (4Max Pro2.0, Shenzhen Anycubic, China). In order to focus on the effect of GnPs, 3D printing process parameters such as layer thickness, feed rate, fill pattern, fill percentage or temperature were fixed for the all the samples. Table 1 shows the values of these parameters.

Table 1. 3D printer parameters.

Parameters

Value

Material

TPU-GnPS

Nozzle diameter

Layer thickness

Printing speed

Nozzle temperature

Bed Temperature

Top/Bottom solid layers

Outline/perimeters shell

Internal fill pattern

External fill pattern

Internal fill percentage

Filament diameter

0.4 mm

0.1 mm

40 mm/s

190 °C

60 °C

1.2 mm

1.2 mm

Mesh

Rectilinear

20%

1.75 mm

----------------------------------------------------------------------------------------

  1. The drawings are too small and its quality need to be improved.

----------------------------------------------------------------------------------------

We apologize for the mistake. As suggested, we have edited the drawings that were not sufficiently clear in the original manuscript.

----------------------------------------------------------------------------------------

  1. Recommendations to present the results of metrological measurements in the form of a table with average value, standard deviation etc and providing the parameters of production.

----------------------------------------------------------------------------------------

We thank the referee for raising this point. The dimensional accuracy and surface roughness are presented in form of a table with average value, standard deviation. Dimensional accuracy is the dimensional variation in length between the FDM printed parts and the CAD dimensions, which is measured using a digital Vernier caliper. The average surface roughness (Ra) of the top surface of FDM parts are measured with a surface roughness tester (TR150A, Timech, China), at 1 mm/s tracing speed, tracing length at 6 mm.

Table 2. The average, standard deviation values of dimensional accuracy and surface roughness.

Samples

Average

Standard deviation

Dimensional accuracy (%)

TPU/GnPs

TPU/IGnPs

TPU/PGnPs

TPU/CGnPs

2.52

2.12

2.58

1.96

0.015

0.108

0.077

0.081

Surface roughness (µm)

TPU/GnPs

TPU/IGnPs

TPU/PGnPs

TPU/CGnPs

1.98

1.91

1.78

1.82

0.384

0.233

0.265

0.252

----------------------------------------------------------------------------------------

  1. It seems that applications should be supplemented with analysis of the influence of processing into roughness, waviness or microstructure, if possible.

----------------------------------------------------------------------------------------

Thanks for the kind suggestion. Because the experimental conditions are limited, a portable surface roughness tester (TR150A, Timech, China) is available in our faculty, only average surface roughness (Ra) is examined and included in the Table 2. The results show that all specimens have high roughness values due to the nature of the additive manufacturing process.

----------------------------------------------------------------------------------------

Reviewer 2 Report

The missing point is why authors were interested in GnP rather than the GO. Since GnP is hydrophobic, it can not be used for 3D printing at all. Obviously polymer additives has to be added. In such scenario, GO/TPU and modifying the structure by post-treatment could be the better option. Authors should discuss this issue and compare the result with the existing reports on GO/TPU 

Generally, electrical conductivity of graphene is reduces with the addition of functional groups. What happened to the modified TPU/GnP such that the electrical conductivity is higher than their pristine counterpart?

The functional groups attached on the surface of graphene acts as a bridging element for the heteroatoms (https://doi.org/10.1016/j.compscitech.2021.109199). Authors should brief on it.

In figure 2b, functional groups attached on the carbon surface are hardly visible. Secondly, the functional groups attached on the surface by different activations methods are different (one is N-free and another GnP/TPU has N-functional groups). Mention the source of N-functional groups and the role of it on the specific properties measured in this study. 

Author Response

The missing point is why authors were interested in GnP rather than the GO. Since GnP is hydrophobic, it can not be used for 3D printing at all. Obviously polymer additives has to be added. In such scenario, GO/TPU and modifying the structure by post-treatment could be the better option. Authors should discuss this issue and compare the result with the existing reports on GO/TPU 

----------------------------------------------------------------------------------------

We sincerely appreciate the comments of the referee and his/her very careful reading of our manuscript.

GnPs consist of multiple layers of graphene corresponding to partially exfoliated graphite. Compared with graphene oxide (GO) and other graphene related materials, GnPs can be easily produced in large scale with physical methods and commercially available with different particle sizes at a relatively low cost. GnPs exhibit exciting properties such as light weight, high aspect ratio, electrical and thermal conductivity, mechanical toughness. GnPs are being widely used in polymer nanocomposites as it showed immense improvement in terms of strength as compared to reduced graphene oxide (rGO), graphene oxide (GO), and multi-walled carbon nanotubes (MWCNT). As a result, GnPs are regarded as a promising carbon nanofiller for nanocomposites due to their good balance of properties and cost.[1]

As evident from the literature review, nowadays many attempts are being taken to develop functional properties using GnPs/polymers in 3D printing. Masarra and Batistella et al.[2] fabricated electrically conductive circuit using PLA/PLC/GnPs by FDM. Misra and Ostadhossein et al.[3] constructed multidrug-eluting stent using direct 3D printing from polycaprolactone/GnPs biodegradable composites. Jing and Chen et al.[4] prepared high thermal conductive polyethylene/GnPs nanocomposites for heat diffusion application of some advanced electronic devices. Li et al.[5] FDM printed wearable pressure sensor based on TPU/GnPs nanocomposites.

We are extremely sorry that any literatures that report GO incorporate only with TPU have not been found, although the widely application of GO in 3D printing.

----------------------------------------------------------------------------------------

Generally, electrical conductivity of graphene is reduces with the addition of functional groups. What happened to the modified TPU/GnP such that the electrical conductivity is higher than their pristine counterpart?

The functional groups attached on the surface of graphene acts as a bridging element for the heteroatoms (https://doi.org/10.1016/j.compscitech.2021.109199). Authors should brief on it.

----------------------------------------------------------------------------------------

GO is a highly exfoliated and functionalized form of graphene. Because of the functionalization, the conjugation within the chemical structure of graphene breaks down in many places, as a result, the density of charge carriers as well as their mobility is decreased. Different from GO, the functionalized GnPs is slightly oxidized GnPs. The high electrical conductivity of TPU/modified GnPs can be attributed through achieving uniform dispersion and distribution of the fillers within TPUs. Cho[6] demonstrated also that the achieving high filler dispersion in a polymer composite is very important for efficiently improving electrical conductivity. The edge-selectively functionalized graphene nanoplatelets (EFG) homogenous dispersed in polyamide 6 matrix can cause a significant improvement of electrical conductivity than the pristine GnPs owing to the electron tunneling effect.[7, 8]

----------------------------------------------------------------------------------------

In figure 2b, functional groups attached on the carbon surface are hardly visible. Secondly, the functional groups attached on the surface by different activations methods are different (one is N-free and another GnP/TPU has N-functional groups). Mention the source of N-functional groups and the role of it on the specific properties measured in this study. 

----------------------------------------------------------------------------------------

We apologize for the mistake. As suggested, we have edited the drawings that were not sufficiently clear in the original manuscript.

The source of the N-functional groups is based on the formation of a zwitterion interaction between the amine or carboxylic acid group on one end of the 12-aminododecanoic acid (ADA) and the GnPs surface. Conversely, there is a strong interaction between the aliphatic chain of the ADD and the TPU matrix due to the organophilic absorption, which leads to the improved modified GnPs distribution. The source of N-free is produced by the reaction between the active species induced by the plasma in the gas phase and the GnPs surface atoms.[9] The plasma contains positive and negative ions, electron, emits UV, among others, that have sufficient enough energy to break chemical bonds at the surface and then create radicals in substrate by reaction with nitrogen, oxygen and thus producing various functional polar groups. By enhancement of the functional groups on the surface of GnPs, the dispersion of GnPs in TPU matrix and the bonding between TPU and matrix will increase.[10]

----------------------------------------------------------------------------------------

References

[1] G. Zhang, F. Wang, J. Dai, Z. Huang, Effect of functionalization of graphene nanoplatelets on the mechanical and thermal properties of silicone rubber composites, Materials 9 (2016) 92.

[2] N.-A. Masarra, M. Batistella, J.-C. Quantin, A. Regazzi, M.F. Pucci, R. El Hage, et al., Fabrication of PLA/PCL/Graphene Nanoplatelet (GNP) Electrically Conductive Circuit Using the Fused Filament Fabrication (FFF) 3D Printing Technique, Materials 15 (2022) 762.

[3] S.K. Misra, F. Ostadhossein, R. Babu, J. Kus, D. Tankasala, A. Sutrisno, et al., 3Dprinted multidrugeluting stent from graphenenanoplateletdoped biodegradable polymer composite, Adv. Healthcare Mater. 6 (2017) 1700008.

[4] J. Jing, Y. Chen, S. Shi, L. Yang, P. Lambin, Facile and scalable fabrication of highly thermal conductive polyethylene/graphene nanocomposites by combining solid-state shear milling and FDM 3D-printing aligning methods, Chem. Eng. J. 402 (2020) 126218.

[5] Z. Li, B. Li, B. Chen, J. Zhang, Y. Li, 3D printed graphene/polyurethane wearable pressure sensor for motion fitness monitoring, Nanotechnology 32 (2021) 395503.

[6] J. Cho, H. Lee, K.-H. Nam, H. Yeo, C.-M. Yang, D.G. Seong, et al., Enhanced electrical conductivity of polymer nanocomposite based on edge-selectively functionalized graphene nanoplatelets, Compos. Sci. Technol. 189 (2020) 108001.

[7] M. Albozahid, H.Z. Naji, Z.K. Alobad, A. Saiani, TPU nanocomposites tailored by graphene nanoplatelets: the investigation of dispersion approaches and annealing treatment on thermal and mechanical properties, Polym. Bull. (2021) 1-39.

[8] A. Paydayesh, S.R. Mousavi, S. Estaji, H.A. Khonakdar, M.A. Nozarinya, Functionalized graphene nanoplatelets/poly (lactic acid)/chitosan nanocomposites: Mechanical, biodegradability, and electrical conductivity properties, Polym. Compos. 43 (2022) 411-421.

[9] A. Rashidi, S. Shahidi, M. Ghoranneviss, S. Dalalsharifi, J. Wiener, Effect of plasma on the zeta potential of cotton fabrics, Plasma Sci. Technol 15 (2013) 455.

[10] M. Baghery Borooj, A. Mousavi Shoushtari, E. Nosratian Sabet, A. Haji, Influence of oxygen plasma treatment parameters on the properties of carbon fiber, J. Adhes. Sci. Technol. 30 (2016) 2372-2382.

Round 2

Reviewer 2 Report

Authors gave satisfactory answer to the queries. No further comments on the updated manuscript.